# Fault-Diagnosis and Fault-Recovery System of Hall Sensors in Brushless DC Motor Based on Neural Networks [note 1]

**DOI:** 10.3390/s23094330

**Published:** 2023-04-27

**Authors:** Kenny Sau Kang Chu, Kuew Wai Chew, Yoong Choon Chang

**Affiliations:** Lee Kong Chian Faculty of Engineering and Science, Universiti Tunku Abdul Rahman, Jalan Sungai Long, Bandar Sungai Long, Kajang 43000, Malaysiaycchang@utar.edu.my (Y.C.C.)

**Keywords:** Hall sensor, fault-detection system, signal-recovery system, CNNLSTM, neural network, brushless direct current motor (BLDC)

## Abstract

This paper proposes a neural-network-based framework using Convolutional Neural Network and Long-Short Term Memory (CNN-LSTM) for detecting faults and recovering signals from Hall sensors in brushless DC motors. Hall sensors are critical components in determining the position and speed of motors, and faults in these sensors can disrupt their normal operation. Traditional fault-diagnosis methods, such as state-sensitive and transition-sensitive approaches, and fault-recovery methods, such as vector tracking observer, have been widely used in the industry but can be inflexible when applied to different models. The proposed fault diagnosis using the CNN-LSTM model was trained on the signal sequences of Hall sensors and can effectively distinguish between normal and faulty signals, achieving an accuracy of the fault-diagnosis system of around 99.3% for identifying the type of fault. Additionally, the proposed fault recovery using the CNN-LSTM model was trained on the signal sequences of Hall sensors and the output of the fault-detection system, achieving an efficiency of determining the position of the phase in the sequence of the Hall sensor signal at around 97%. This work has three main contributions: (1) a CNN-LSTM neural network structure is proposed to be implemented in both the fault-diagnosis and fault-recovery systems for efficient learning and feature extraction from the Hall sensor data. (2) The proposed fault-diagnosis system is equipped with a sensitive and accurate fault-diagnosis system that can achieve an accuracy exceeding 98%. (3) The proposed fault-recovery system is capable of recovering the position in the sequence states of the Hall sensors, achieving an accuracy of 95% or higher.

## 1. Introduction

The paper focuses on a Brushless DC (BLDC) motor that employs three internal Hall sensors strategically positioned 120 degrees apart within the motor, resulting in a 60-degree resolution configuration [1,2,3,4,5]. Hall sensors serve as fundamental components in various types of motors and play a vital role in measuring the motor’s speed and position [6]. The Hall sensors’ output signals serve as inputs to control the motor by computing its speed and position. Consequently, the controller can leverage these input signals to generate a precise pulse width modulation signal (PWM) to the driver, allowing for effective control of the motor system. Detecting faults and recovering the system’s signal using a fault-diagnosis and -recovery system for Hall sensors is crucial. In state-of-the-art fault-diagnosis and -recovery systems, Ref. [1] utilized the switching sequence of the motor as a reference and the average value of the Hall sensor signals as an algorithm to check the health of the sensors. Then a remedial measure method was used as the fault-recovery system, which reconstructed the signals. In [2], a quasi-rotating space vector was employed to represent the Hall-effect sensor states in a stationary reference frame. When faults were detected, and the space vector deviated from normal operation, the algorithms were utilized for fault compensation to predict the rotor position and speed. In [4], a state-sensitive method was proposed as the fault-diagnosis method, and the fault-recovery method determined the lost transition and transition instant prediction to recover the signal. On the other hand, in [7,8], a sensorless algorithm had been proposed to use a predictive model in order to predict the αβ-axis current for controlling the speed of the motor.

Machine learning, particularly deep learning, has been widely adopted in various fields, such as data analysis and engineering, due to its ability to handle a large amount of data and solve complex tasks, such as image and text recognition. In recent years, machine learning technology has rapidly advanced and been applied in many areas, including fault diagnosis in motors [9]. In reference [10] the authors implemented a search tree algorithm in the motor control system to enhance the model’s robustness by predicting the voltage vector. Fault diagnosis in motors is a classification problem that requires a labeling method to distinguish different types of faults. For example, reference [11] describes the use of machine learning techniques for fault diagnosis in motors. Unfortunately, with the increase in data complexity and size, it can be challenging for single-model neural networks, such as the CNN model or RNN model, to effectively learn and extract features from the data. To overcome this limitation, the hybrid CNN-LSTM model has been proposed and implemented in various classification and estimation tasks. The CNN model can extract information from spatial data, while the LSTM can extract temporal dependencies in sequential data. This results in more accurate and robust models for complex data tasks. For instance, reference [12] proposed a hybrid CNN-LSTM model for forecasting particulate matter (PM2.5), which is composed of complex data. Their research shows promising results, as their model achieved the lowest mean absolute error (MAE) and root mean square error (RMSE). Similarly, reference [13] proposed a hybrid CNN-BiLSTM neural network model for sentiment classification based on text-based data from social media. Their model achieved the highest accuracy as compared to other neural network models. Ref. [14] proposed a CNN-LSTM model for quench detection, where the data consist of quench voltage signals with noise. The CNN model extracts features, and the LSTM model captures temporal dependencies in the data. Furthermore, Ref. [15] developed a CNN-LSTM model with a joint regional correlation threshold denoising (WRCTD) algorithm for the fault-detection system of the harmonic reducer of the motor system. The data include torque/RPM sensors and acceleration sensor readings, and the CNN-LSTM model achieved the highest accuracy compared to other neural network models. In a previous study regarding the Hall sensor fault-diagnosis system [16] a neural network model was successfully employed to detect faults in Hall sensors with an accuracy exceeding 98%. Signal recovery systems play a vital role in restoring faulty signals, allowing the motor system to regain its normal function. The present study aims to develop a CNN-LSTM model to train both the fault-diagnosis and fault-recovery systems, while also combining the fault-diagnosis system from previous research with the signal-recovery system. The contributions of this study are as follows: (1) CNN-LSTM neural network structure is proposed to be implemented in both the fault-diagnosis and fault-recovery systems for efficient learning and feature extraction from Hall sensors data. (2) The proposed fault-diagnosis system is equipped with a sensitive and accurate fault-diagnosis system that can achieve accuracy exceeding 98%. (3) The proposed fault-recovery system is capable of recovering the position in the sequence states of the Hall sensors, achieving an accuracy of 95% or higher.

## 2. Materials and Methods

According to the arrangement of the Hall sensors, its result is a 60-degree resolution arrangement [1,2,3,4,5]. The Hall sensors’ signals are a crucial factor in controlling the performance of the motor system, as controllers require precise and accurate information on the motor’s position or speed to compute switching signals for the driver, ultimately maintaining the motor system’s efficiency. In this context, fault-diagnosis systems of Hall sensors have played an important role in detecting faults in the system. The utilization of a fault-diagnosis system is aimed at preventing malfunctions and promoting the longevity of the system. Conventionally, the equation-based models produced estimated output to indicate the faults taking place in the system. This approach has a limitation on implementing the method to different types of motors since it uses a specific equation to design the model.

Machine learning has been extensively adopted in many fields such as data analysis (big data), engineering (estimation modeling and filtering), etc. [17]. Deep learning is a sub-section of machine learning [18]. It is powerful and scalable, and has the ability to take in a large amount of data and solve complex tasks such as image recognition, text recognition, and others. In recent years, machine learning technology has been evolving rapidly, and has been used in many applications. It has a good performance in pattern recognition. Machine learning techniques are able to extract patterns and learn from the raw data. For instance, fault diagnosis uses a convolutional neural network (CNN) to detect faults that occur in the bearings and the gears of the motor [19]. Identification of the faults in the motor is an example of a classification problem and a labeling method is applied for the machine learning model to distinguish different kinds of faults [11].

### 2.1. Fault Diagnosis in Hall Sensor

Brushless DC (BLDC) motors are widely used in various applications due to their high efficiency and reliability [20]. These motors consist of three internal Hall sensors (H1, H2, and H3) that output signals used by the controller to control the speed and position of the motor. The output signals from the Hall sensors are typically in digital format, represented as either “0” or “1” [2,21]. However, faults can occur in the Hall sensors, which can have a negative impact on the performance of the motor. In order to detect and diagnose such faults, a fault-diagnosis system is applied to the system. A current fault-diagnosis model has been proposed that utilizes an estimation model and sequence method for fault detection. This allows the system to identify and classify various types of faults that may occur in the Hall sensors. Several algorithms have been proposed for fault diagnosis in BLDC motors, such as the Vector Tracking Observer (VTO). The VTO algorithm compares the output signal of the VTO with the estimated output signal in order to detect any changes or differences, which may indicate the presence of a fault [22]. Other techniques, such as the Fast Fourier Transform (FFT), have also been used to filter the signal in order to identify faults [23]. By implementing these algorithms, the fault-diagnosis system is able to accurately and reliably detect and classify faults in the Hall sensors of a BLDC motor. This allows for timely maintenance and repair to be performed, ensuring the optimal performance of the motor.

The sequence method for fault detection involves following a specific sequence or pattern and comparing it to the output signals from the Hall sensors [16]. There are three main approaches to this method: (1) The signal-state-sensitive method involves storing a few previous states in the controller for comparison. If the current state does not match the expected sequence or pattern, a fault is detected. (2) The Hall sensors transition order approach involves storing the latest two or more transitions in the controller memory. If the output transition sequence for a damaged Hall sensor is different from the normal operation transition sequence, a fault is detected. (3) The condition-based method uses algorithms and conditions to generate a set of sequences, and a comparison and tracking function is used to observe any changes. If any changes are detected, a fault is indicated [4,24].

Table 1 shows the commutation of the motor with Hall sensor states under normal operation. If faults occur in the Hall sensors, the fault-diagnosis system can use algorithms or compare the sequences to detect the faults that have occurred. This is achieved by comparing the expected output of the Hall sensors based on their characteristic behavior under normal operation (as shown in Table 1) with the actual output of the sensors. Any deviations from the expected output can be used to identify and classify faults in the Hall sensors.

### 2.2. Type of Faults

Table 2 lists the digital signals generated by the Hall sensors and the corresponding machine learning state labeling for various conditions [16]. For a BLDC motor with three Hall sensors, there are three main types of faults that can occur: (1) Single fault (Type I), (2) Double fault (Type II), and (3) Triple fault (Type III). The fault-diagnosis system can use these labels to classify and identify the specific type of fault that has occurred in the Hall sensors.

Faults can have a variety of causes, such as damage to the sensor itself or problems with the sensor’s connection to the controller. A single fault is a fault that only occurs in one of the Hall sensors. Figure 1 shows an example of a single fault that occurred in a Hall sensor. A double fault occurs when two sensors are experiencing issues. Figure 2 shows an example of a double fault in which Hall sensors 2 and 3 remained in the state ‘0’, indicating that there were faults present in these sensors. A triple fault occurs when all three Hall sensors in the BLDC motor are experiencing issues and fail to operate under normal conditions. Figure 3 shows the behavior of a triple fault case.

### 2.3. Hall Sensor Fault-Recovering System

The state-of-the-art model of the signal-recovery system is based on an algorithm to estimate the rotor position and angular speed of the motor using the last correct signal from the Hall sensor. In reference [1], when the fault was detected, the sensor signal was substituted with a computed signal from a fictitious Hall sensor by the controller. This procedure was referred to as a remedial measure. In [2,25], three models were used: (1) the zeroth-order algorithm, (2) the hybrid observer, and (3) the Vector Tracking observer to determine the estimated rotor speed and position. In [3], the transition instant prediction was used to determine the estimated speed and rotor position. First, the correct timings of Hall sensor signal transitions (determined in the fault-diagnosis system) were determined, and the last correct Hall sensor signal was recorded. By using the correct transition and the last correct signal, the speed and rotor position were predicted. In [23], a Luenberger observer was applied to the mechanical dynamic model to determine the estimated rotor position. In the proposed fault-recovery system, the model used three Hall sensors and the fault-detection output to compute the correct Hall sensor signal.

#### Working Principle of Proposed Fault-Recovery System

According to Table 1, the number of sequences in the Hall sensor signal was fixed at 6 phases, as the sensors were detected in 60 resolutions, resulting in 6 phases per oscillation. A neural network model was used to determine the motor’s position, which required finding the phases in the sequence of the Hall sensor signal. During the training process of the fault-recovery system’s neural network, two types of inputs were required: (1) Hall sensor signal and (2) fault-diagnosis system’s output. The model was trained with the three Hall sensors’ signals and the output of the fault-diagnosis system, enabling it to extract features of the three Hall sensors’ signals with different outputs from the fault-detection system. Figure 4 depicts the block diagram of the proposed fault-recovery system, where the three Hall sensors’ signals and fault-diagnosis system’s output passed through the neural network to detect the specific phases in which the motor was positioned. The output generated the correct Hall sensors’ signal based on the output from both neural networks.

## 3. Methodology

CNN-LSTM Neural Network hybrid setup is selected for the choice of the proposed model.

### 3.1. Proposed Hall Sensor Fault-Diagnosis System and Fault-Recovering System

In this research, several machine learning models were implemented to create a fault-detection system and signal-recovery system for a brushless DC (BLDC) motor that has three internal Hall sensors spaced 120 degrees apart. Deep neural networks (DNNs) are known to perform well in pattern recognition tasks, but it is also important to have a sufficient and diverse dataset to train a high-performing fault-detection system model. It is necessary to include all types of faults in the training data so that the model can learn to handle various scenarios that may occur in the system. To train the fault-recovery system which requires Hall sensors’ signal with the fault-detection system’s output. Different conditions of the fault-diagnosis system’s output gave different sets of sequences of Hall sensors’ signals. In order to train a good fault-recovery system, it required an adaptive model to extract more information from the complex data.

The proposed model was developed using the TensorFlow and Keras libraries. The inputs are first processed by a convolutional layer (CNN) which uses a convolutional operation to obtain feature mapping. The convolutional kernel slides through the input and calculates the integral of the pointwise multiplication based on the size of the input. The output of the convolutional layer is then used as the input for the LSTM layer, but a dense layer is added between the CNN output and the LSTM layer to extract information through the time steps and input size. Finally, the LSTM’s output is processed by an output layer (dense) that uses an activation function such as the SoftMax function to compute probability distributions for each class, and the class with the highest probability is chosen.

The convolutional neural network (CNN) model, as depicted in Figure 5, is a type of neural network that is particularly well-suited for tasks involving the analysis of two-dimensional data such as images. One key aspect of CNNs is the use of convolutional layers, which apply a mathematical operation called convolution to the input data. Convolution involves sliding a kernel, or small matrix, over the input data and calculating the integral of the elementwise product between the kernel and the input. This process allows the CNN to identify patterns and features in the data that are relevant for the task at hand. The output of the convolutional layer is a set of feature maps that capture different aspects of the input data. The equation for the convolutional layer can be expressed as:(1)$$yc_{i,j}=f\left({\left(W*X\right)}_{i,j}+b_{m}\right)$$
The feature map is produced by applying a convolution operation to the input (*X*) and weight (*W*), and adding the bias constant ($b_{m}$). The row (*i*), column (*j*), and layer (*m*) of the feature map are denoted by variables *i*, *j*, and m, respectively. The output of this operation is then standardized using a non-linear function (*f*) as described in references [26,27]. After the feature map is extracted, it is passed through a pooling layer which shrinks the input and reduces computational load and memory usage. The pooling layer also helps to prevent overfitting. Finally, the input is classified using a fully connected layer and an activation function (e.g., SoftMax, ReLu, etc.). The SoftMax function layer calculates the probability of the input data belonging to the machine learning state labeled class [26,28]. The SoftMax function is frequently employed as the activation function in a multi-class classifier’s output layer, with *K* denoting the number of classes. The function is defined as follows:(2)$$\sigma {\left(z\right)}_{i}=\frac{e^{zi}}{\sum_{j=1}^{K}e^{zj}}$$
where the SoftMax function (σ) is utilized to determine the probability that input data (*z*) corresponds to each class. This is achieved through the exponential function (*e*), which divides the exponential of the input ($e^{zi}$) by the sum of the exponentials of the outputs ($e^{zj}$) based on the index *j* and upper limit *K*.

Recurrent neural networks (RNNs) are designed to process sequential data such as time series or natural language. They have a memory input called the hidden state (represented by ht in Figure 6a) which allows them to incorporate information from past time steps in the input sequence. This is in contrast to traditional feedforward neural networks, which process one input at a time and do not incorporate past information. RNNs are trained using backpropagation, a process in which the network receives an input, produces an output, and adjusts its weights and biases to reduce the error between the output and the desired output. The equation of RNN can be expressed as follows:(3)$$y_{t}=W_{y}\times h_{t}$$
The RNN output, denoted by *y*${}_{t}$, is associated with the weight *W${}_{y}$*, which serves to incorporate the RNN’s memory, represented by *h${}_{t}$*, into the output. The memory formula is precisely defined as follows:(4)$$h_{t}=f\left(W_{h}\times h_{t-1}+W_{x}\times x_{t}+b\right)$$
The memory of the RNN at time step *t* is calculated using the non-linear activation function (*f*), the weights *W*${}_{h}$ and *W${}_{c}$*, the memory from the previous time step (*h${}_{t-1}$*), the input at the current time step (*x${}_{t}$*), and the bias constant (*b*).

One limitation of the recurrent neural network (RNN) model is its difficulty in training on long sequences of data [29]. This can result in vanishing gradients, where the gradients of the weights in the network become very small and thus have a minimal impact on the network’s output, or exploding gradients, where the gradients become very large, leading to unstable training. To address these issues, the Long Short-Term Memory (LSTM) model was introduced as an improvement on the RNN model. The layout of an LSTM is illustrated in Figure 6b. In contrast to the RNN, which has only one type of memory, the LSTM has both short-term and long-term memory, allowing it to capture both short-term and long-term dependencies in the data. This enables the LSTM to effectively handle long sequences and maintain information from earlier time steps, as it can selectively store and forget information in its short-term and long-term memory cells. The LSTM is composed of four functions: (1) Forget: the equation for the forget function is expressed as
(5)$$F=\sigma (h_{t-1})$$
in which *F* is the output of the forget function and σ is the sigmoid function. (2) Store: the equation of the store function is defined as
(6)$$S=\left(F\times S_{t-1}\right)+\left(\sigma \left(h_{t-1}\right)\times tanh\left(h_{t-1}\right)\right)$$
in which *S* is the output of the store function, * is element-wise multiplication, *S*${}_{t-1}$ is the output of the store function from the previous time step, and tanh is the hyperbolic tangent function. (3) Update: the equation of the store function is defined as
(7)$$U=U_{t-1}\times F+S$$
in which *U* is the output of the update function and *U*${}_{t-1}$ is the output of the update function from the previous time step. (4) Output: the equation of the LSTM output function is defined as
(8)$$Y=h_{t}=F\times tanh\left(S\right)$$
in which output (*Y*) of the LSTM is produced using four functions: forget, store, update, and output [27,30]. The forget function discards irrelevant information from the previous time step, while the store function saves pertinent new information in the cell state. The update function then selectively updates the information in the cell state, and the output function controls the information that is passed on to the next time step.

There are several different machine learning algorithms. The CNN-LSTM neural network hybrid setup was chosen for the proposed model for the beneficial reasons listed. The 1D-CNN layer setup is selected for the CNN model layer as the CNN model can learn the characteristics of the raw data through the convolutional and pooling layer. Subsequently, the LSTM model is to serve the purpose of identifying the illustration of the sequential data and the model is specifically designed to learn to recognize crucial input and store it in a long state. The combination of CNN and LSTM hybrid model setup offers the benefit of better feature-extraction ability and improves the robustness of the model [31,32]. The CNN-LSTM hybrid setup as shown in Figure 7 mainly consists of an input layer, 1D-CNN layer, LSTM layer, and output layer.

### 3.2. Procedure of the Proposed Model and Experimental Arrangement

The motor specification is shown in Table 3. The experimental setup for the fault-diagnosis and fault-recovery system is presented in Figure 8. The Hall sensor signals served as inputs to the controller, which then sent the signals to the processor containing the fault-diagnosis and signal-recovery neural networks. The processor computed the correct Hall sensor signals, which were then fed back to the controller for generating PWM signals to control the BLDC motor. The experimental setup, as shown in Figure 9, consisted of a 24 V, 84 W BLDC motor and a three-phase MOSFET-based H-bridge driver connected to a DC power supply. The experiments were conducted using the Python programming language, which involved loading the model files and Hall sensor signals from the driver into the program for execution. The fault-detection model file was used to classify the types of faults present in the Hall sensor signals, while the signal-recovery model file was used to recover the correct signals for the system to ensure smooth functioning of the system.

The proposed model was implemented according to the flowchart presented in Figure 10. All trained machine learning models were saved in an h5 file format and loaded into the program for fault-diagnosis and fault-recovery systems. Upon initialization of all parameters, the motor was started and readings from all Hall sensors were simultaneously obtained from the motor driver. These readings were updated in a CSV file and extracted for use as input for the fault-detection model. If no fault was detected, the program sent the Hall sensors’ signal to the controller to compute PWM for the driver to control the motor. However, if a fault was detected, the fault-detection system model identified the type of fault and affected sensors. The fault Hall sensors’ signal and the output of the fault-diagnosis system were sent to the signal-recovery system for recovery of the actual Hall sensors’ signal. The output of the fault-recovery system was then sent to the output process to compute the actual Hall sensors’ signal form. The corrected Hall sensors’ signal was sent to the controller to generate PWM for the driver to control the BLDC motor. Finally, the program was reset, and the loop continued.

## 4. Results and Discussion

In the research, the topic was mainly divided into two main parts: (1) fault-diagnosis system and (2) fault-recovery system. In this research, a neural network was developed to detect faults in a BLDC motor with three internal Hall sensors. The model was implemented using the TensorFlow and Keras libraries, and it included a convolutional layer for feature extraction, a long short-term memory (LSTM) layer for extracting information through time steps and input size, and an output layer with a SoftMax activation function for classifying the states and generating the machine learning output [30].

### 4.1. Fault-Diagnosis Neural Network

The model was trained using Google Colab and tested on a BLDC motor setup with a 1D-convolutional layer, a MAXpooling layer, a Dense layer, and a Flatten layer. The model’s layer and parameter are shown in Table 4. The results showed that the model was able to accurately detect faults in the Hall sensors and identify the type of the fault.

#### 4.1.1. Performance of the Proposed Fault-Diagnosis System

The confusion matrix graph is used to determine both the performance of the fault-diagnosis method and the accuracy of the fault-detection system in the nine different machine learning state labels. The performance of the CNN-LSTM model setup is represented in Figure 11. The fault-detection system was proven to achieve an accuracy of 98% in nine different states. The performances of the state sensitive method and transition-sensitive method are shown in Figure 12 and Figure 13, respectively. Although the two methods have slightly better performance in determining the normal operation than the CNN-LSTM model, the two methods have biases in normal operating conditions. The confusion matrices of the state-sensitive method and transition-sensitive method have illustrated some probability of the other eight conditions being categorized in the normal operation (label as ‘0’).

The performances of different types of methods are encapsulated in Figure 14. By comparison between all the methods, the CNN-LSTM model can accurately detect the fault, and the performance of the CNN-LSTM model is higher than other methods. The SS method and TS method have the issue under some scenarios that their output is the same as the normal operation. Hence, they are unable to determine whether any faults occur in the system. Figure 15 shows that, in a single Hall sensor fault scenario (H2 sensor has a fault), both the SS method and TS method have delays in detecting the fault. It is because the SS and TS methods require taking in a few transition signals to compute the output. For instance, Figure 15 shows that the SS method and TS method take three and five transition signals respectively. Figure 16 depicts a double Hall sensor fault case (H1 and H2 have faults). The performance of the SS method is poorer than the CNN-LSTM model and TS method, due to the SS method having some conditions which are the same in other class labelings. For instance, it has mixed up a few No. 5 class labeling conditions with No. 8 class labeling. The proposed model has the most stable result among the three methods and the model has a higher overall performance than the state sensitive method and transition sensitive method.

#### 4.1.2. Performance of the Different Types of Machine Learning Models in Proposed Fault-Detection System

The findings from the experimental evaluation of five distinct neural network models utilized for fault detection in BLDC motors are presented in Table 5, while Table 6 outlines the structural components and parameters of the various neural networks in the fault-diagnosis system. Notably, only three of the models were able to successfully detect all three types of faults. The CNN-LSTM model demonstrated the most stability and efficiency out of the five, combining the advantages of both CNN and LSTM. Although its training duration was slightly longer than that of the CNN model, it was still shorter than that of the LSTM model. Conversely, the DNN model exhibited the shortest training duration, which can be attributed to its straightforward architecture. However, both the RNN and LSTM models required the longest training durations due to the increased computation of weights and memory inputs in their equations. These models were unable to differentiate between the various types of faults and could only predict the presence or absence of a fault. This can be attributed to the design of RNN and LSTM structures, which are intended to handle time-based or sequential data, and are therefore only capable of classifying data into two states. All the neural networks were trained using the Adam optimizer with a learning rate of 0.001, a batch size of 20, and 500 epochs were set. An early stopping function with a patience of 10 was implemented to prevent overfitting.

### 4.2. Fault-Recovery Neural Network

In the research, the neural network was designed to identify the position of the motor which is the phase in the sequence of the Hall sensors signal. CNN-LSTM was chosen to be used as the neural network for the system.

#### 4.2.1. Performance of Fault-Recovery System in Neural Network Model

The confusion matrix of the signal-recovery system is presented in Figure 17. The results show an overall average accuracy of 98% across all six different phases. It is worth noting that the Phase 3 scenario experienced signal overlap with Phase 0 and Phase 5, resulting in lower accuracy for these two phases. This was due to some cases having the same inputs as other phases. In the event of a triple-fault scenario, the neural network was unable to accurately determine the phases as all of the sensor signals were incorrect. However, under the condition where at least one of the sensor signals was correct, the neural network was able to predict the phase accurately.

#### 4.2.2. Performance of Different Neural Network in Fault-Recovery System

This section describes the evaluation of five different neural network models in the signal-recovery system. Table 7 summarizes the performance of each model, with RNN and LSTM being excluded due to incompatibility issues. The accuracy of CNN, CNN-LSTM, and DNN were comparable, all averaging around 97%, while the CNN-LSTM model achieved the highest accuracy. During training, the CNN-LSTM model required approximately 13–14 min, which was intermediate compared to the other models. To optimize the neural network model, the number of hidden layers was a crucial factor to consider. Table 8 illustrated the accuracy of the different numbers of hidden layers in the CNN-LSTM model. As shown in Table 8, the CNN-LSTM 3 achieved the highest accuracy and was thus selected as the optimal model for training in the fault-recovery system. All neural networks in our fault-recovery system were trained using the Adam optimizer with a learning rate of 0.001 and a batch size of 32. Table 9 shows the comparison between the architectures in different types of neural network. The input layer of each network had a size of 50, while the hidden layer utilized the ReLU as the activation function. The output layer was constructed with six nodes and the Softmax activation function to accommodate the six distinct states in the Hall sensor signal sequence.

#### 4.2.3. Performance of Fault-Recovery System

Figure 18 depicts the graph showing the actual Hall sensors signal, the recovered Hall sensors signal, and the output of the fault-diagnosis and fault-recovery system under a single fault condition (H1). Meanwhile, Figure 19 illustrates the graph of speed versus time under the same fault condition. The experiment triggered a fault at 0.0142 s, which was diagnosed by the system and generated an output of 2 at 0.0144 s, indicating that H1 was faulty. The fault-recovery system was able to generate signals at 0.0145 s, thereby reducing the speed by 150 rpm. The speed returned to the reference value after 2.8 ms of duration.

Figure 20 displays a graph illustrating the actual Hall sensors signal, the recovered Hall sensors signal, and the output of the fault-diagnosis and fault-recovery system under a double fault condition (H1, H2). Similarly, Figure 21 depicts a graph showing the speed versus time under the same fault condition. During the experiment, a fault was triggered at 0.272 s, resulting in an output of 5 from the fault-diagnosis at 0.0275 s, indicating that both H1 and H2 were faulty. The fault-recovery system generated signals at 0.0280 s, leading to a reduction in speed by approximately 175 rpm. The longer recovery time required for the double fault scenario led to a more significant drop in speed compared to the single fault scenario. The correct signal took 0.008 s to generate. The speed returned to the reference value after a duration of 3 ms.

Figure 22 depicts a graph showing the actual Hall sensors signal, the recovered Hall sensors signal, and the output of the fault-diagnosis and fault-recovery system under triple fault condition (H1, H2, H3). Likewise, Figure 23 displays the graph of speed versus time under the same fault condition. During the experiment, a fault was triggered at 0.04 s, resulting in outputs of 6 and 8 from the fault diagnosis at 0.0402 s. These outputs indicated that H1, H2, and H3 were faulty. The reason for output 6 was that the original signal of H2 was 0. Output 8 was generated when the system detected that H2 could not function. The fault-recovery system was able to generate signals at 0.041 s, causing a reduction in speed by approximately 175 rpm. It took 3 ms for the speed to return to the reference value. In the case that all Hall sensors fail at the onset, the system cannot recover the signal due to the absence of any viable benchmark signals.

## 5. Conclusions

In conclusion, the convolutional long short-term memory (CNN-LSTM) neural network architecture has proven to be a successful approach for implementing both the Hall sensor fault-diagnosis system and fault-recovery system. By training the machine learning model on the Hall sensors signal data, it has overcome the limitations of traditional fault-detection methods by learning patterns from the data and being more adaptive in detecting faults. In the fault-recovery system, the neural network effectively computed the motor’s position and generated accurate signals for the controller. The combination of CNN and LSTM has significantly improved the information extraction and the models’ robustness. The experimental results have demonstrated that the proposed model achieved high accuracy and good performance. This deep learning approach offers higher flexibility and accuracy, making it a powerful tool for the motor Hall sensor fault-diagnosis system and fault-recovery system. Overall, the CNN-LSTM model has shown outstanding performance in this field, highlighting the potential of deep learning in this domain.

## Figures and Tables

**Figure 1 sensors-23-04330-f001:**
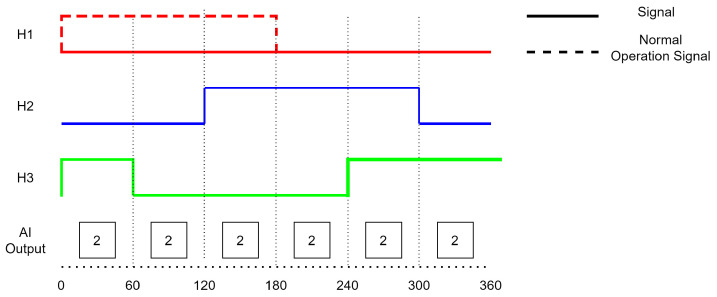
Single Fault (H1).

**Figure 2 sensors-23-04330-f002:**
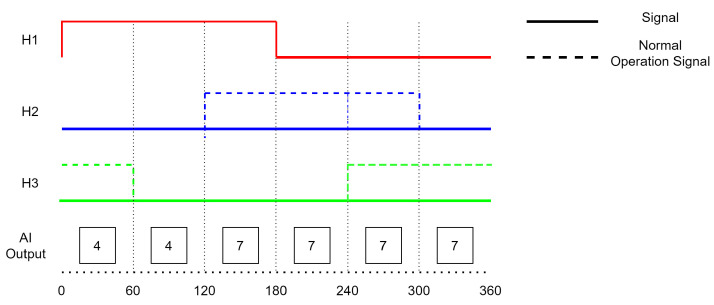
Double Fault (H2, H3).

**Figure 3 sensors-23-04330-f003:**
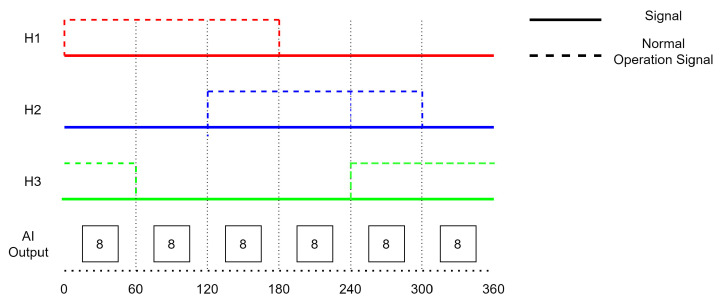
Triple Fault (H1, H2, and H3).

**Figure 4 sensors-23-04330-f004:**
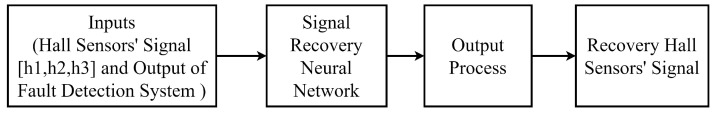
Block Diagram of the Proposed Signal Recovery System.

**Figure 5 sensors-23-04330-f005:**
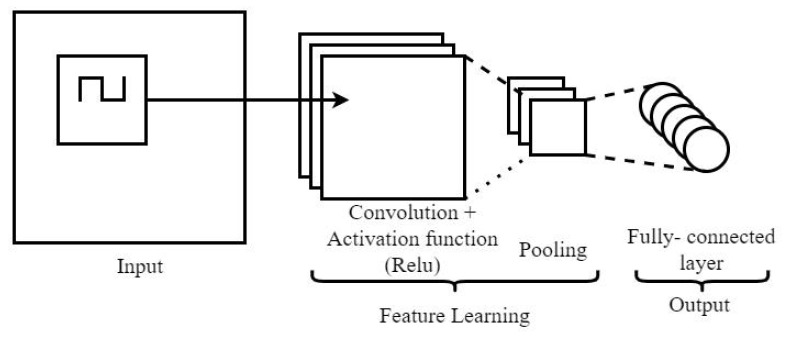
Layout of a CNN Model.

**Figure 6 sensors-23-04330-f006:**
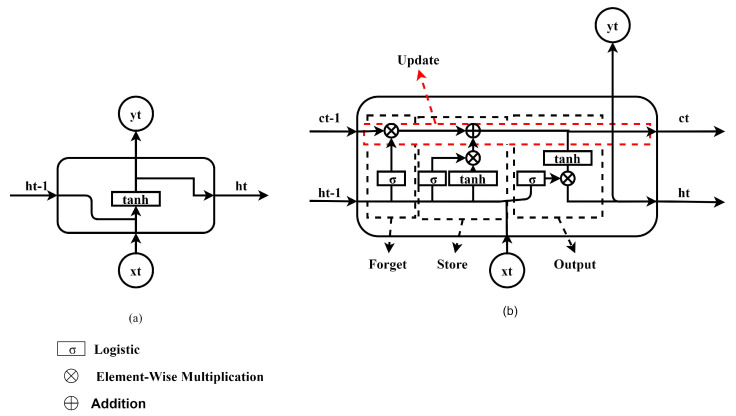
(**a**) Internal Structure of RNN; (**b**) LSTM Model.

**Figure 7 sensors-23-04330-f007:**
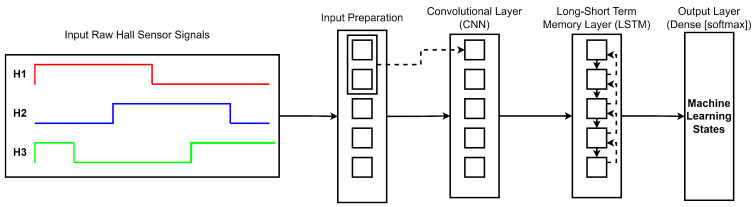
Structure of the Proposed Machine Learning Model.

**Figure 8 sensors-23-04330-f008:**
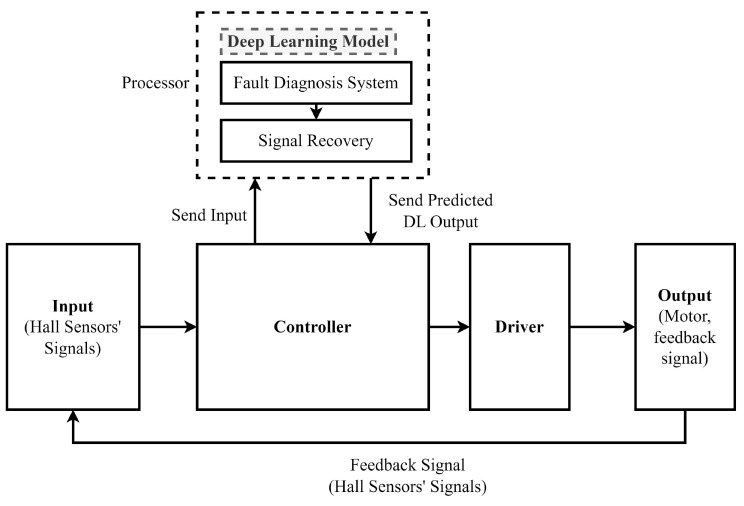
Block Diagram of Detailed Experiment Setup.

**Figure 9 sensors-23-04330-f009:**
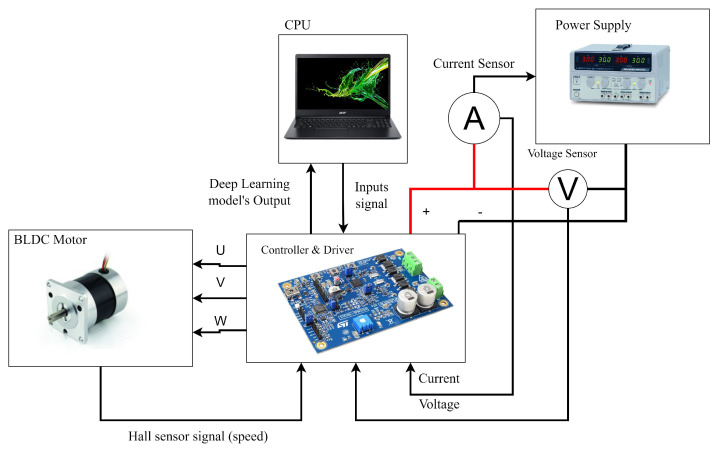
Experiment Setup.

**Figure 10 sensors-23-04330-f010:**
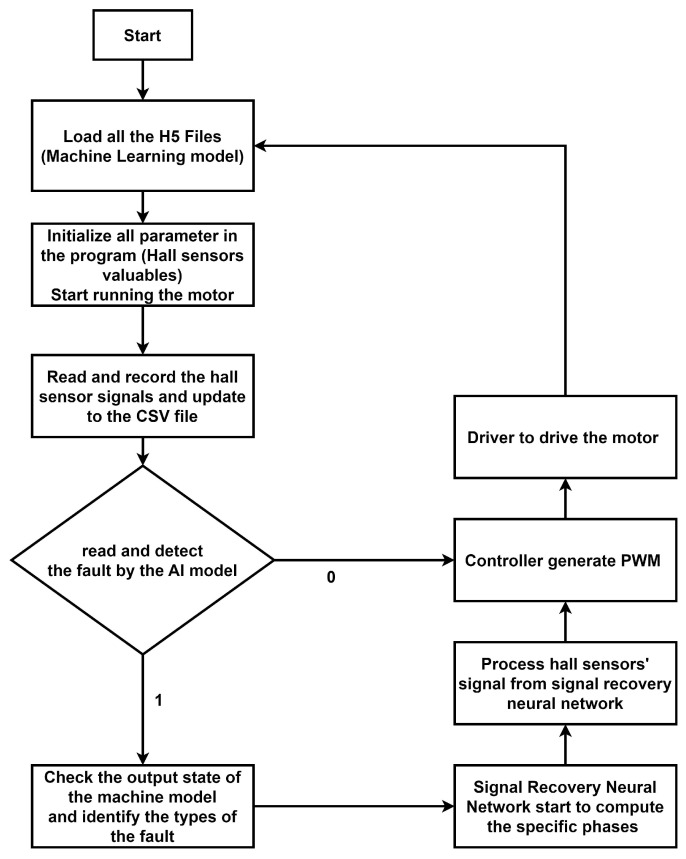
Flow Chart of the System.

**Figure 11 sensors-23-04330-f011:**
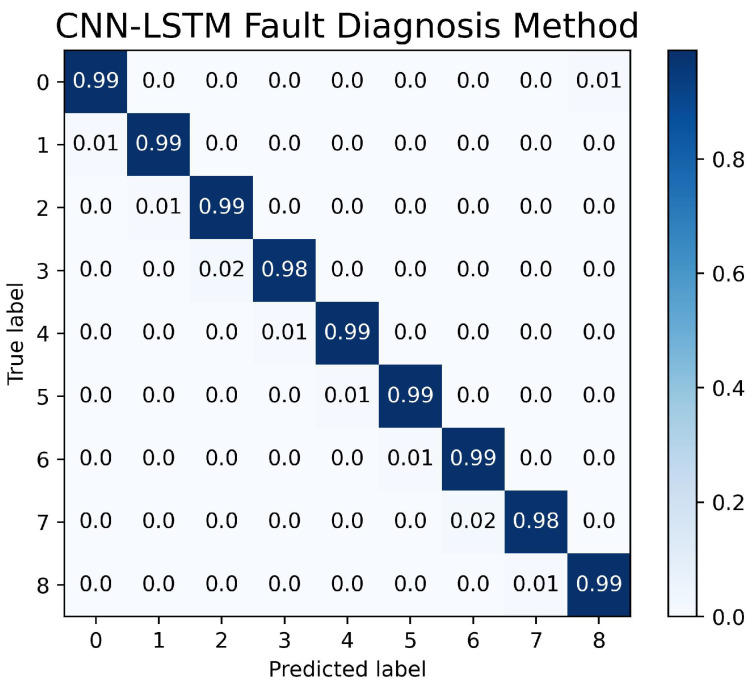
Confusion Matrix of CNN-LSTM Fault-Detection System.

**Figure 12 sensors-23-04330-f012:**
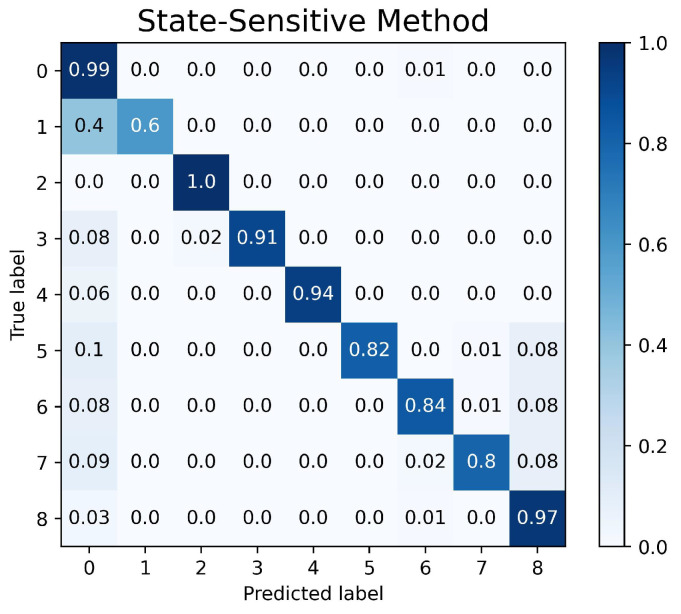
Confusion Matrix of State Sensitive Method.

**Figure 13 sensors-23-04330-f013:**
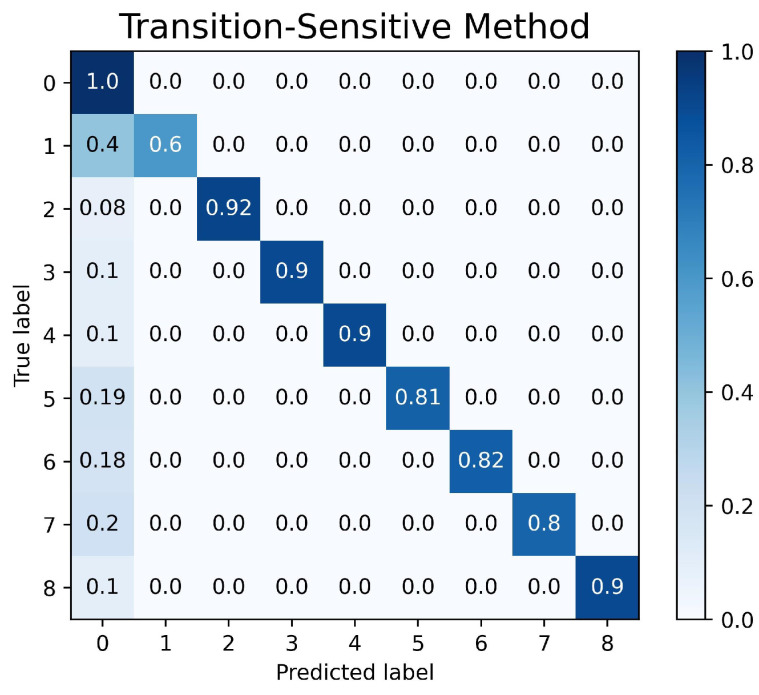
Confusion Matrix of Transition Sensitive Method.

**Figure 14 sensors-23-04330-f014:**
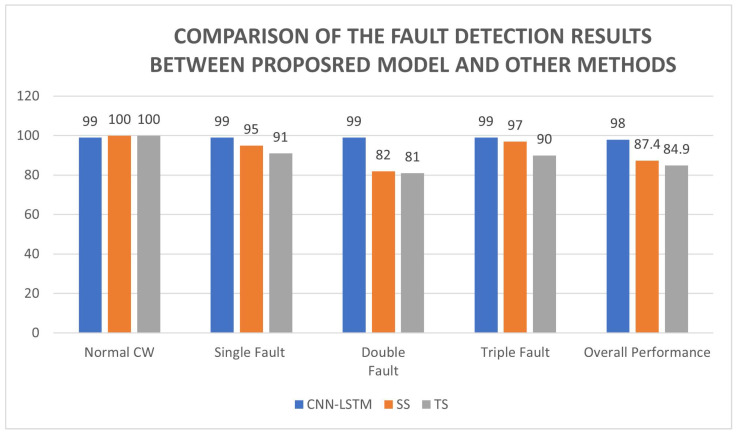
Comparison of the Fault-Detection Results Between Proposed and the Other Methods.

**Figure 15 sensors-23-04330-f015:**
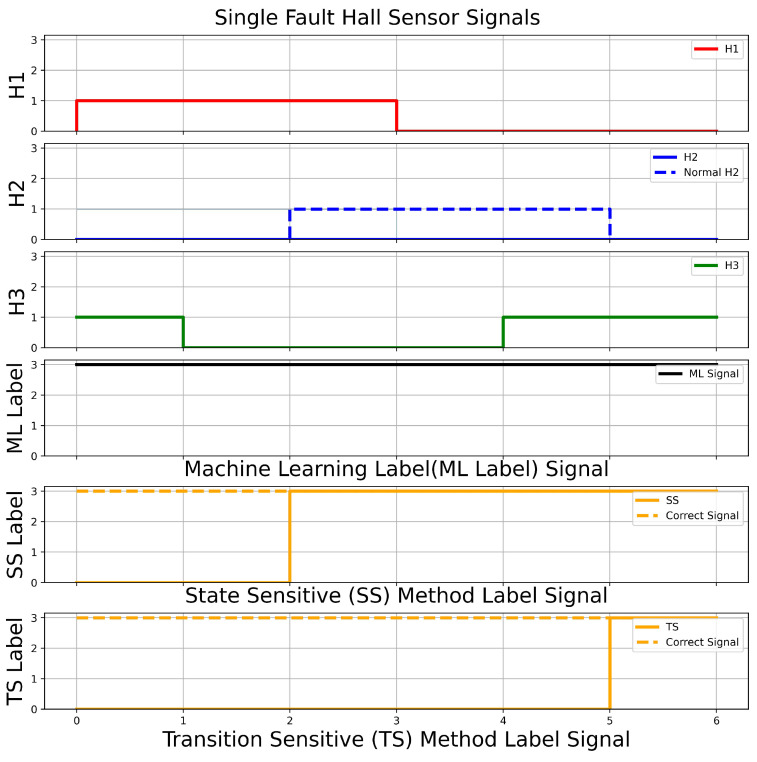
Single Fault Case (H2) in Comparison Between Different Methods.

**Figure 16 sensors-23-04330-f016:**
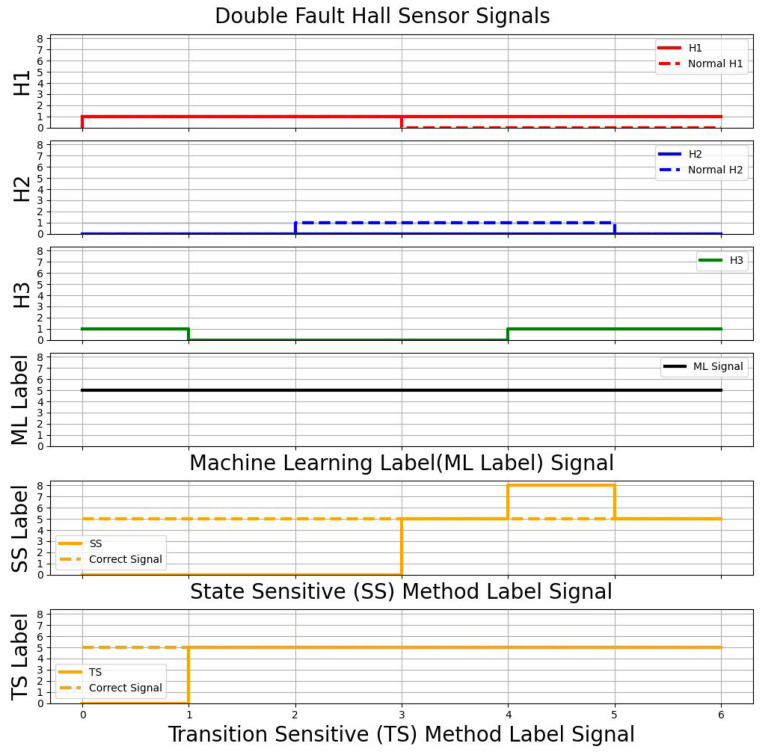
Double Fault Case (H1, H2) in Comparison Between Different Methods.

**Figure 17 sensors-23-04330-f017:**
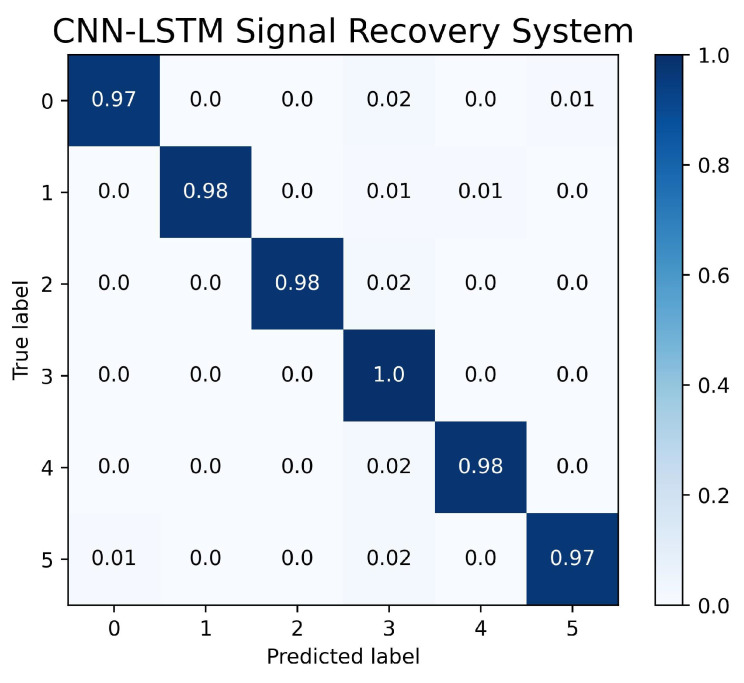
Confusion Matrix of CNN-LSTM Signal Recovery System.

**Figure 18 sensors-23-04330-f018:**
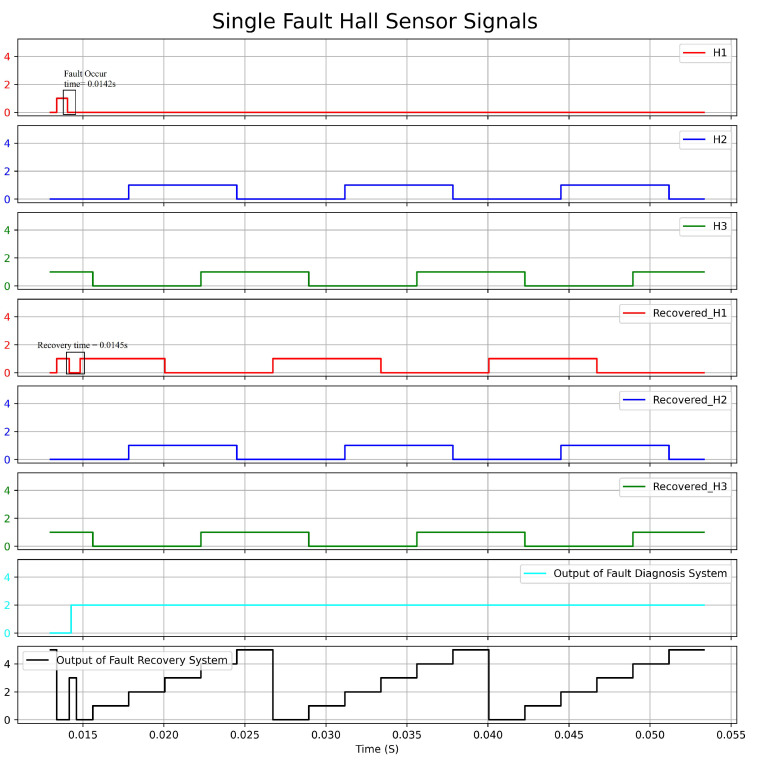
Response Graph of the Hall Sensors Signal and the Output of Fault Diagnosis and Fault Recovery under Single Fault Condition.

**Figure 19 sensors-23-04330-f019:**
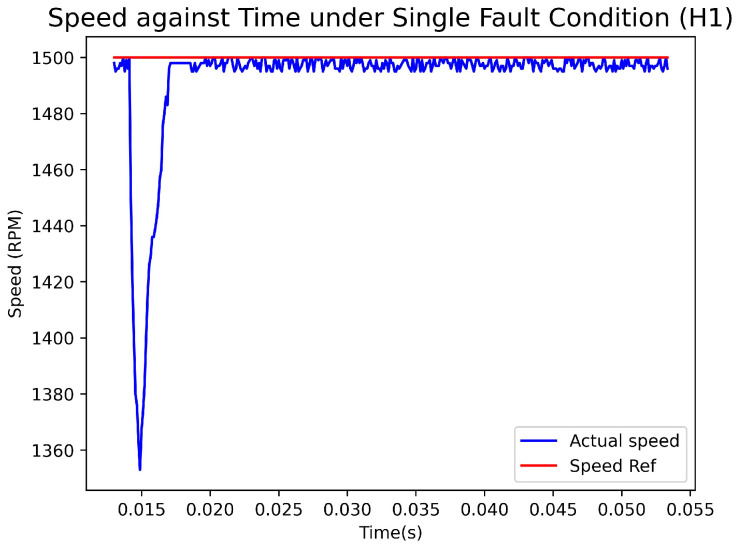
Speed(rpm)–Time(s) Graph with Single Fault condition.

**Figure 20 sensors-23-04330-f020:**
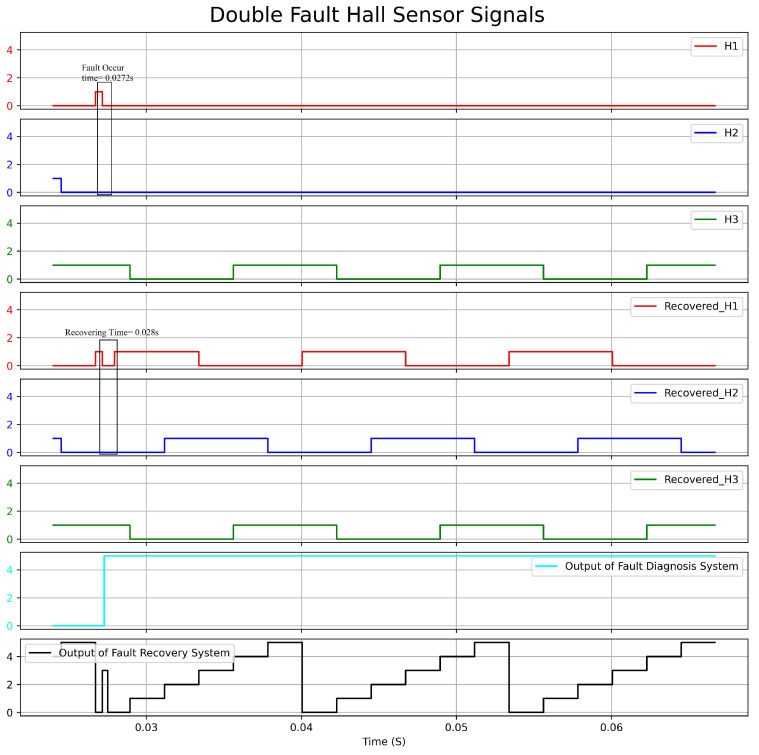
Response Graph of the Hall Sensors Signal and the Output of Fault Diagnosis and Fault Recovery under Double Fault Condition.

**Figure 21 sensors-23-04330-f021:**
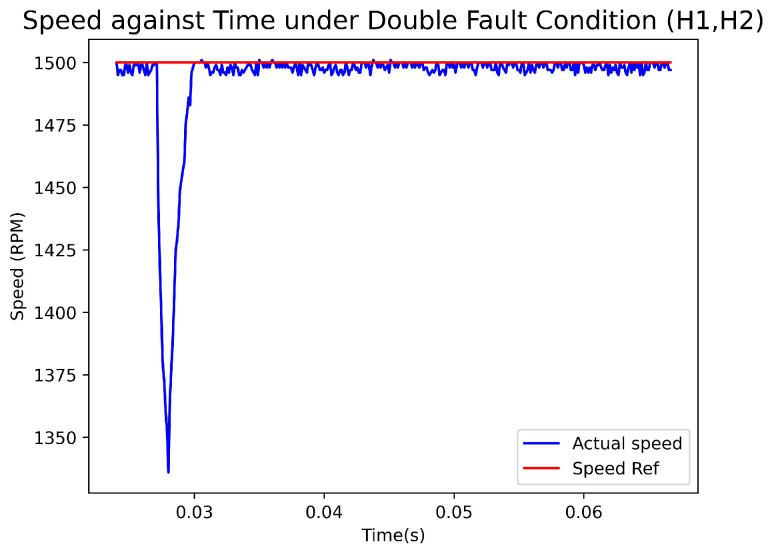
Speed(rpm)–Time(s) Graph with Double Fault condition.

**Figure 22 sensors-23-04330-f022:**
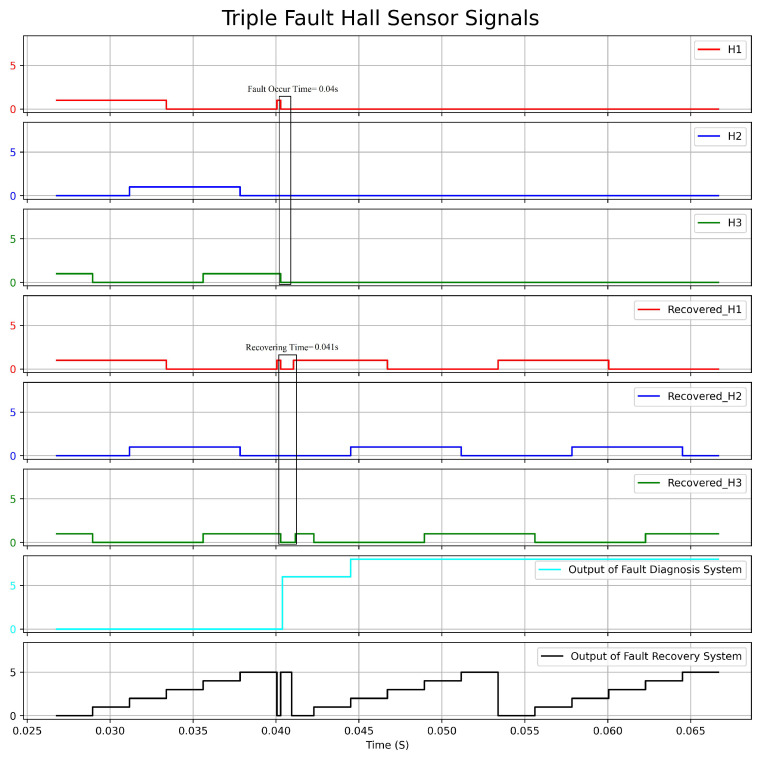
Response Graph of the Hall Sensors Signal and the Output of Fault Diagnosis and Fault Recovery under Triple Fault Condition.

**Figure 23 sensors-23-04330-f023:**
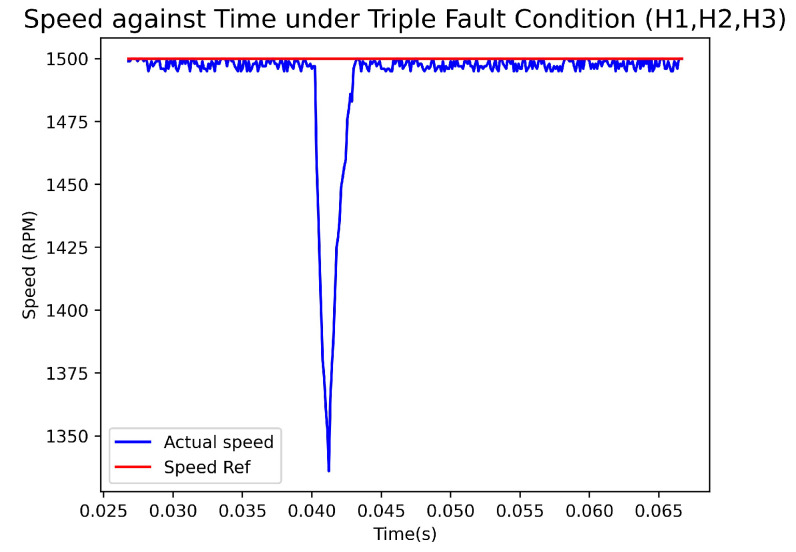
Speed(rpm)–Time(s) Graph with Triple Fault condition.

**Table 1 sensors-23-04330-t001:** Properties of normal Hall sensor signal output.

			Hall Sensor	
Angle of the Motor	Phases in Sequence of the Hall Sensor Signals	H1	H2	H3
0–60°	0	1	0	1
60–120°	1	1	0	0
120–180°	2	1	1	0
180–240°	3	0	1	0
240–300°	4	0	1	1
300–360°	5	0	0	1

**Table 2 sensors-23-04330-t002:** Fault Cases in Machine Learning States.

	Sensor with Fault	State Label
Normal Signal (Clockwise Direction)	None	0
Normal Signal (Anti-clockwise Direction)	None	1
Single fault (Type I)	H1	2
	H2	3
	H3	4
Double fault (Type II)	H1, H2	5
	H1, H3	6
	H2, H3	7
Triple fault (Type III)	H1, H2, H3	8

**Table 3 sensors-23-04330-t003:** Electrical Specification of the BLDC Motor.

Parameter	Value
Rated Voltage (V)	24
Rated Current (A)	5
Rated Power (W)	84
Rated Torque (Nm)	0.23
Rated Speed (rpm)	3500
No. of Poles	6

**Table 4 sensors-23-04330-t004:** Parameter of CNN-LSTM Model.

Layer	Output Shape
Conv1D	(None, 46, 4)
Dense	(None, 46, 50)
Dense	(None, 46, 50)
LSTM	(None, 50)
Flatten	(None, 50)
Dense	(None, 9)

**Table 5 sensors-23-04330-t005:** Overall Performance of Different Types of Neural Networks in Fault-Diagnosis System.

Type of Neural Network	Variety of Faults Detected	Training Period (minutes)	Data Size	Efficiency Fault (%)
DNN	1, 2, 3	3–5	5400	99.1–99.3
CNN	1, 2, 3	10–13	5400	99.0–99.3
RNN	1	45–60	5400	None
LSTM	1	45–60	5400	None
CNN-LSTM	1, 2, 3	13–15	5400	99.3–99.4

**Table 6 sensors-23-04330-t006:** Comparison of Layers Designed in Different Types of Neural Networks in Fault-Diagnosis System.

Layer	DNN	CNN	RNN	LSTM	CNN-LSTM
1	Dense(50, ReLU)	Conv1D(50, 4, ReLU)	RNN(50, tanh)	LSTM(50, tanh)	Conv1D(50, 4, ReLU)
2	Flatten	Max Pool-1D(2)	Dense(50, ReLU)	Dense(50, ReLU)	Max Pool-1D(2)
3	Dense(50, ReLU)	Dense(25, ReLU)	Dense(9, softmax)	Dense(9, softmax)	Dense(50, ReLU)
4	Dense(25,ReLU)	Flatten	-	-	Dense(50, ReLU)
5	Dense(25, ReLU)	Dense(9, softmax)	-	-	LSTM(50, tanh)
6	Dense(9, softmax)	-	-	-	Flatten
7	-	-	-	-	Dense(9, softmax)

**Table 7 sensors-23-04330-t007:** Overall Performance of Different Types of Neural Networks in Fault-Recovery System.

Type of Neural Network	Training Period Detect	Data Size (minutes)	Efficiency (%)
DNN	3–5	5400	96.5–97.0
CNN	10–12	5400	96.9–97.0
RNN	45–60	5400	None
LSTM	45–60	5400	None
CNN-LSTM	13–14	5400	97.1–97.3

**Table 8 sensors-23-04330-t008:** Comparison of Number of Hidden and the Accuracy of CNN-LSTM Model in Fault-Recovery System.

	CNNLSTM 1	CNNLSTM 2	CNNLSTM 3	CNNLSTM 4	CNNLSTM 5
	Conv1D	Conv1D	Conv1D	Conv1D	Conv1D
	MaxPooling	MaxPooling	MaxPooling	MaxPooling	MaxPooling
	1D	1D	1D	1D	1D
	LSTM	Dense	Dense	Dense	Dense
	Flatten	LSTM	Dense	Dense	Dense
	Dense	Flatten	LSTM	Dense	Dense
	-	Dense	Flatten	LSTM	Dense
	-	-	Dense	Flatten	LSTM
	-	-	-	Dense	Flatten
	-	-	-	-	Dense
Accuracy(%)	96.8	97.0	97.2	96.9	96.5

**Table 9 sensors-23-04330-t009:** Comparison Between Different Types of Neural Networks’ Architecture in Fault-Recovering System.

Layer	DNN	CNN	RNN	LSTM	CNN-LSTM
1	Dense(60, ReLU)	Conv1D(60, 4, ReLU)	RNN(60, tanh)	LSTM(60, tanh)	Conv1D(60, 4, ReLU)
2	Flatten	Max Pool-1D(2)	Dense(60, ReLU)	Dense(60, ReLU)	Max Pool-1D(2)
3	Dense(60, ReLU)	Conv1D(30, 4, ReLU)	Dense(6, softmax)	Dense(6, softmax)	Dense(30, ReLU)
4	Dense(30,ReLU)	Max Pool-1D(2)	-	-	Dense(30, ReLU)
5	Dense(30, ReLU)	Dense(18, ReLU)	-	-	LSTM(30, tanh)
6	Dense(12, ReLU)	Flatten	-	-	Flatten
7	Dense(6, softmax)	Dense(6, softmax)	-	-	Dense(6, softmax)

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
