# Peer review of "Fault-Diagnosis and Fault-Recovery System of Hall Sensors in Brushless DC Motor Based on Neural Networks†"

_sensors, 2023, doi:10.3390/s23094330_

Round 1

Reviewer 1 Report

This paper used a CNN-LSTM neural network for fault detection and faulty signal recovery of hall sensors in brushless DC motors. CNN and LSTM are both standard neural network methods – what the novelty elements are in this paper and what the original contributions are should be made clearer.

It is stated that the proposed method is simple and flexible – can you explain how it is simple, e.g., with computational time comparison, and how it is flexible, again with measurable indicators for the flexibility of the implementation or the usage of the model.

It is stated with an efficiency of 99.3% or 97% -- is this supposed be classification accuracy?

Figure 6 can be made clearer.

Introduction is very short, which should be extended to include critical review of literature in fault diagnosis, faulty signal recovery and deep learning methods, e.g., intelligent fault diagnosis of rotor-bearing system under varying working conditions with modified transfer CNN and thermal images; fault diagnosis of asynchronous motors based on LSTM neural network, etc. In addition to the novelty elements and contributions, why CNN and LSTM need to be combined could also be justified in Introduction. 

The baseline techniques for comparison in Table 4 and Table 5 could be explained more in detail, e.g., their architecture and hyperparameters. The fairness of the comparisons should also be discussed.

Author Response

Thank you for taking the time to review my paper. Your feedback is very helpful and I appreciate your comments. 

Reviewer 2 Report

please see attached file.

Author Response

Thank you for taking the time to review our paper. Your feedback is greatly appreciated and has been incredibly helpful in improving the quality of our work.
